# Roles of Bile Acid-Activated Receptors in Monocytes-Macrophages and Dendritic Cells

**DOI:** 10.3390/cells14120920

**Published:** 2025-06-18

**Authors:** Huilin Jia, Xingli He, Tengfei Jiang, Fanzhi Kong

**Affiliations:** College of Animal Science and Veterinary Medicine, Heilongjiang Bayi Agricultural University, No. 5 Xinfeng Road, Sartu District, Daqing 163319, China; huilinjia@byau.edu.cn (H.J.); hxl98747500@163.com (X.H.); tengfeijiang207@byau.edu.cn (T.J.)

**Keywords:** bile acids, farnesoid X receptor, G protein-coupled bile acid receptor 1, liver X receptors, vitamin D receptor, monocytes-macrophages, dendritic cells

## Abstract

Bile acids (BAs), essential for lipid metabolism and fat-soluble vitamin absorption, also act as signaling molecules that regulate immune homeostasis. This review focuses on the roles of four key BA-activated receptors, farnesoid X receptor (FXR), G protein-coupled bile acid receptor 1 (GPBAR1), liver X receptors (LXRs), and vitamin D receptor (VDR), in modulating the functions of monocytes-macrophages, and dendritic cells (DCs). The biological synthesis, transport, and metabolism of BAs were discussed and highlighted the feedback mechanisms regulating the synthesis and enterohepatic circulation of BAs. Each receptor’s role in shaping immune responses is detailed, including their function in inflammation, apoptosis, phagocytosis, and pathogen clearance. FXR and GPBAR1 activation generally exhibits anti-inflammatory effects, while LXR and VDR modulate a more nuanced interplay between immune responses and lipid homeostasis. We also explored the cross-talk between BA-activated receptors and Toll-like receptors, providing a comprehensive understanding of the complex interplay between BA signaling and innate immunity. This review culminates by highlighting the therapeutic potential of targeting these receptors for the treatment of inflammatory and autoimmune diseases.

## 1. Introduction

Bile acids (BAs) are amphipathic molecules derived from cholesterol, characterized by their hydrophilic and hydrophobic properties, which render them essential components in the mammalian digestive system. These molecules play a pivotal role in lipid metabolism, facilitating the absorption of lipids and fat-soluble vitamins [1]. Clinically, BAs are employed in treating liver and biliary disorders, including gallstones and primary biliary cholangitis [2,3]. Recent research has demonstrated that BAs can inhibit the replication of hepatitis viruses [4] and coronaviruses within the intestines [5]. Additionally, they act as vital signaling molecules, regulating glucose and lipid metabolism, energy homeostasis, and suppressing inflammation within immune cells via receptor activation [6]. The key receptors involved include the farnesoid X receptor (FXR), G protein-coupled bile acid receptor 1 (GPBAR1, also known as TGR5), liver X receptors (LXRs), and the vitamin D receptor (VDR). These receptors orchestrate the functions of innate immune cells such as monocytes, macrophages, and dendritic cells (DCs), playing critical roles in maintaining innate immune homeostasis. This article elucidates the mechanisms by which bile acid receptors FXR, GPBAR1, LXR, and VDR modulate monocyte-macrophage and dendritic cell functions, offering insights that may facilitate the development of immunomodulatory therapeutics targeting these receptors.

## 2. Biological Synthesis, Transport, and Metabolism of BAs

BAs are synthesized in liver cells through cholesterol oxidation mediated by cytochrome P450 enzymes via two distinct pathways: the classical pathway and the alternative pathway [7] (Figure 1). Within the liver, primary bile acids (pBAs), namely cholic acid (CA) and chenodeoxycholic acid (CDCA), are derived from cholesterol through a series of enzyme-catalyzed reactions. Post-synthesis, CA and CDCA undergo conjugation with taurine or glycine to form taurocholic acid (TCA), taurochenodeoxycholic acid (TCDCA), glycocholic acid (GCA), or glycochenodeoxycholic acid (GCDCA) [7]. Concurrently, pBAs are secreted from the liver into bile canaliculi. These BAs can either be directly secreted into the duodenum or stored and concentrated in the gallbladder. The presence of acidic and fatty chyme in the intestine stimulates the release of cholecystokinin from intestinal endocrine type I cells into the systemic circulation, prompting gallbladder contraction and the subsequent release of bile into the duodenum.

In the intestines, microbial enzymes from the gut microbiota metabolize pBAs [8]. Primary unconjugated BAs like CA and CDCA are transformed into secondary unconjugated BAs, specifically deoxycholic acid (DCA), lithocholic acid (LCA), and ursodeoxycholic acid (UDCA), through processes such as 7α-dehydroxylation or 7β-hydroxyl epimerization. These secondary unconjugated BAs can be conjugated with taurine or glycine to form glyco- or tauro-conjugated BAs, including glyco/tauro-deoxycholic acid (G(T)DCA), glyco/tauro-lithocholic acid (G(T)LCA), or glyco/tauro-ursodeoxycholic acid (G(T)UDCA) [9].

Primary conjugated BAs (GCA, TCA, GCDCA, and TCDCA) can be deconjugated into CA and CDCA by bile salt hydrolases (BSH). Recent studies have shown that BSH exhibits N-acyltransferase activity, allowing it to bind amines with BAs to form bacterial bile acid amidates (BBAAs) [10]. In organoid models, BBAAs can activate host ligand-activated transcription factors, such as the pregnane X receptor (PXR) and the aryl hydrocarbon receptor (AHR) [10].

At the end of the ileum, approximately 95% of intestinal BAs are reabsorbed into intestinal epithelial cells via the bile salt export pump (BSEP) and apical sodium-dependent bile acid transporter (ASBT), and then they secreted into the portal vein through basolateral BAs transport proteins, including the organic solute transporter α/β (OSTα/β) subunit complex [11]. Subsequently, BAs are transported into hepatocytes by the sodium/taurocholate co-transporting polypeptide (NTCP) and organic anion-transporting polypeptide 1 (OATP1) [11]. BAs can also enter the systemic circulation through alternative pathways, involving multidrug resistance-associated protein 3 (MRP3), MRP4, and OSTα/β [11]. Furthermore, OATP1A2, OATP1B3, NTCP, and ABST are also expressed on monocyte, macrophages, or DCs and can mediate the uptake of a variety of anionic compounds, including bile acids [12,13,14]. The remaining 5% of BAs are either excreted in feces or utilized by the gut microbiota.

## 3. BA-Activated Receptors

### 3.1. FXR

BA-activated receptors constitute a diverse family of nuclear receptors and G protein-coupled receptors that are either activated or inhibited by BAs, exhibiting notable species specificity [15]. The FXR was initially identified as an orphan nuclear receptor, but subsequent research established its role as a bile acid receptor [16]. Various BAs, including CDCA, LCA, DCA, and CA, can activate FXR, with CDCA being the most potent FXR ligand in humans and CA being the most effective in mice [17]. Interestingly, α-muricholic acid (α-MCA) and β-muricholic acid (β-MCA), two pBAs unique to mice, act as FXR antagonists [18]. Additionally, UDCA is regarded as a neutral or weak FXR antagonist [18].

FXR is widely expressed across various organs, including the liver, heart, kidneys, and intestines [17]. It is found in hepatocytes, cholangiocytes, liver sinusoidal cells, as well as intestinal and liver endothelial cells [19]. FXR expression is also observed in several immune cell types, such as monocytes, macrophages, DCs, natural killer (NK) cells, and NKT cells, although its levels are relatively lower in T cells and B cells [20,21]. When activated by ligands, FXR in different innate immune cells can modulate their functions, thereby contributing to the homeostasis of innate immunity and enhancing defense against both exogenous antigens and those produced by the gut microbiota [16].

### 3.2. LXR

LXRs were initially identified as orphan receptors within the nuclear receptor superfamily. Subsequent studies revealed that oxidized metabolites of cholesterol serve as ligands for LXRs, facilitating their activation [22]. Upon activation, LXRs can induce aerobic glycolysis, acetyl-CoA production, and de novo cholesterol synthesis [23]. There are two homologous subtypes of LXRs: LXRα and LXRβ. LXRα is predominantly expressed in the liver, but it is also found in various tissues, including the intestine, kidneys, lungs, adipose tissue, and macrophages [24]. In contrast, LXRβ is more universally expressed across most tissues [24].

LXRs function as key transcription factors that regulate lipid homeostasis in mammals, modulating the expression of genes involved in cholesterol metabolism [25]. Endogenous ligands for LXRs include oxidized metabolites derived from cholesterol biosynthetic intermediates [22]. The activation of LXRs, together with the signaling pathways of mammalian target of rapamycin (mTOR) and sterol regulatory element-binding protein c1 (SREBP-c1), regulates the increased biosynthesis of fatty acids and cholesterol during the latter stages of toll-like receptor (TLR) responses [23].

Research has shown that LXRα mediates the expression of the liver cholesterol 7α-hydroxylase (Cyp7a) gene, which is induced by dietary cholesterol [26]. Studies involving LXRα knockout mice indicated that feeding LXRα (-/-) mice a diet containing 2% cholesterol significantly increased their liver cholesterol content [26]. Additionally, LXRα (-/-) mice exhibited a reduced BA pool size and lower BA excretion compared to wild-type mice when consuming a high-cholesterol diet [26]. These findings underscore the role of LXRα in cholesterol metabolism and its importance in maintaining cholesterol and BA homeostasis [26]. Furthermore, it has been discovered that 6α-hydroxylated BAs specifically activate LXRα, and while these BAs can also activate LXRβ, a higher concentration is required for LXRβ activation [27].

### 3.3. VDR

Vitamin D is a fat-soluble vitamin with two primary forms: vitamin D2 (ergocalciferol) and vitamin D3 (cholecalciferol) [28]. Vitamin D3 undergoes two hydroxylation steps in the liver and kidneys to yield its active form, 1,25-dihydroxyvitamin D3 (1,25(OH)2D3). The key enzymes involved in this activation are CYP27A1 and CYP27B1 [29]. Notably, the expression of CYP27B1 is not confined to the kidneys; it is also expressed in monocytes, macrophages, and DCs, where it plays an essential intracellular role [29]. Moreover, LCA acts as a ligand for the VDR, and both LCA and its derivatives can activate VDR [30]. VDR is widely expressed in various tissues and organs, including the ileum, kidneys, liver, bones, skin, and endocrine tissues. Its expression is particularly pronounced in non-parenchymal liver cells, such as Kupffer cells (KCs), hematopoietic stem cells, and cholangiocytes [31]. Additionally, VDR is present in immune cells, including T cells, B cells, NK cells, DCs, monocytes, and macrophages [32], thereby contributing to the regulation of immune responses.

### 3.4. GPBAR1

GPBAR1 is a member of the G protein-coupled receptor family, characterized by seven transmembrane domains. Secondary bile acids (sBAs), such as DCA and LCA, are the preferred ligands for GPBAR1 and are present in the highest concentrations at the distal ileum and in the colon [16]. While sBAs are predominant, pBAs, including CDCA and CA, can also activate GPBAR1 [33]. GPBAR1 is expressed in various tissues, including the intestines, stomach, liver, spleen, gallbladder, and adipose tissue [33]. Within the liver, it is localized to sinusoidal endothelial cells, cholangiocytes, and hepatic stellate cells, but it is not expressed in hepatocytes [33]. Similar to FXR, GPBAR1 is also widely present in multiple immune cell types, including monocytes, macrophages, DCs, NK cells, and NKT cells [34]. The roles of various BAs and their receptors in the regulation of gut immunity remain incompletely understood; however, their effects appear to involve both overlapping and independent mechanisms [35]. Notably, studies indicate that intestinal inflammation can spontaneously develop under stable conditions in FXR knockout (FXR-/-) mice that express GPBAR1 [20,21], as well as in GPBAR1 knockout (GPBAR1-/-) mice that express FXR normally [34].

### 3.5. Other BA-Activated Receptors

In addition to the previously discussed receptors, BAs can also activate several other receptors, including the PXR [36], constitutive androstane receptor (CAR) [37], sphingosine 1-phosphate receptor 2 (S1PR2) [38], muscarinic receptors 2 and 3 (M2 and M3) [6], and formyl peptide receptor 1 (FPR1) [39]. PXR is activated by various ligands and plays a critical role in regulating the expression of drug-metabolizing enzymes. Additionally, PXR is involved in governing genes that contribute to the homeostasis and exogenous biotransformation of BAs within the gut-liver axis [36]. CAR, another key nuclear receptor, regulates genes associated with drug metabolism and clearance [37]. It influences the expression of enzymes essential for metabolizing xenobiotics and endogenous compounds, including hormones and BAs [37]. The activation of CAR can affect liver detoxification processes as well as energy metabolism [37]. S1PR2 is implicated in the regulation of cell migration, proliferation, and survival, contributing to vascular development and immune responses [40]. In the context of BAs, S1PR2 has been associated with modulating liver regeneration and fibrosis [41]. M2 primarily play a role in regulating heart rate by slowing cardiac function [42]. They are also present in the central nervous system, where they influence neural signaling [42]. M3 help regulate smooth muscle contraction and glandular secretion and may also affect gastrointestinal motility [43]. These receptors are implicated in airway constriction and can be found in exocrine glands and the gastrointestinal tract [43]. FPR1 is crucial in mediating immune cell responses, particularly in directing leukocyte migration towards sites of infection or inflammation [44]. It serves as a vital component of the innate immune response and plays a role in controlling inflammation [45]. The distribution, natural ligands, and antagonists of BA-activated receptors are summarized in Table 1.

## 4. Roles of BA-Activated Receptors in Monocytes-Macrophages

### 4.1. The Anti-Inflammatory Function of FXR Activation in Monocytes-Macrophages

As previously noted, the FXR and GPBAR1 are prominent BA receptors expressed in circulating monocytes and macrophages derived from the gut and liver. Research indicates that FXR regulates the activity of these immune cells at both the intestinal and hepatic levels [20,21]. Numerous studies have shown that the activation of FXR in macrophages from both humans and rodents can diminish their pro-inflammatory activity. For instance, Renga et al. demonstrated that FXR modulates the immunoregulatory activity of TLR-9, thereby influencing the inflammatory response in intestinal macrophages [48]. The anti-inflammatory effects triggered by FXR ligands are mediated through multiple mechanisms, some of which depend on the atypical nuclear receptor small heterodimer partner (SHP), while others are independent of SHP [48]. Furthermore, Yang et al. found that SHP binds to the C-C motif chemokine ligand 2 (CCL2) promoter, blocking the recruitment of nuclear factor-kappa B (NF-κB) and stabilizing inhibitory complexes that restrict chemokine production [49].

Several studies have reported that aged FXR knockout (FXR-/-) mice exhibit impaired expression of inflammatory mediators and increased intestinal permeability [50,51]. Upon exposure to dextran sulfate sodium (DSS) or 2,4,6-trinitrobenzene sulfonic acid (TNBS), these mice develop severe disease [21,50]. Additionally, severe intestinal inflammation has been observed in human samples from patients with Crohn’s disease and ulcerative colitis, as well as in wild-type mice treated with TNBS or DSS [21]. This inflammation is associated with an increase in NF-κB-dependent cytokines, including tumor necrosis factor-alpha (TNF-α), interleukin-6 (IL-6), IL-1β, and inducible nitric oxide synthase (iNOS), further supporting the link with deletion of FXR expression [21].

Research indicates that treatment of wild-type mice exposed to DSS or TNBS with obeticholic acid (OCA) results in reduced expression of pro-inflammatory marker genes such as IL-1β, IL-6, and monocyte chemoattractant protein-1, confirming the immunoregulatory role of FXR. This regulatory activity involves both NF-κB-dependent and independent pathways [21,52]. When FXR is activated by its ligand, it promotes the expression of certain pro-inflammatory genes, including iNOS and IL-1β. This binding stabilizes the nuclear receptor corepressor 1 (NCoR1) complex, which associates with FXR bound to the NF-κB response element (Figure 2A). This stabilization prevents the kB subunit from directly interacting with the gene’s promoter. In contrast, TLR-4 activation leads to the release of NCoR1 from the promoter, facilitating the transcription of inflammatory genes. In summary, the interplay between FXR and various inflammatory pathways underscores their critical roles in regulating immune responses and maintaining intestinal homeostasis.

### 4.2. LXR Activation Regulates Inflammatory, Apoptotic, Phagocytic, and Pathogen Clearance Functions in Monocytes-Macrophages

In macrophages, LXRs perform multiple essential functions, including the regulation of inflammatory responses, apoptosis, and phagocytosis. Activation of LXRs in mice exhibits anti-inflammatory effects, thereby influencing immune responses favorably [23]. However, in humans, LXR activation appears to limit the anti-inflammatory polarization of macrophages, leading them toward a more pro-inflammatory phenotype [23]. Studies indicate that LXR ligands can suppress the expression of inflammatory factors, such as iNOS, IL-1β, and TNF-α, during bacterial infections or lipopolysaccharide (LPS) stimulation [53,54]. This ligand-mediated suppression of inflammatory gene transcription by LXRs is dependent on the recruitment of NCoR and involves SUMOylation; LXRs are recruited to the iNOS promoter in a SUMO-dependent manner, inhibiting the release of corepressors from inflammatory gene promoters and consequently suppressing pro-inflammatory gene transcription [54]. Furthermore, LXRs can downregulate the expression of pro-inflammatory factors like iNOS and cyclooxygenase-2 (COX-2) by inhibiting NF-κB activity [55].

Reports have shown that LXRα activation by ligands promotes the expression of the nuclear receptor orphan receptor 1 (NOR-1) in KCs, leading to increased IL-10 production and suppression of LPS-induced inflammation [56]. Additionally, LXRs can inhibit LPS-induced IL-18 expression in mouse bone marrow-derived macrophages by suppressing the expression of caspase-1 and the IL-18 precursor [57]. LXRs also enhance the expression of the endogenous IL-18 inhibitor, IL-18 binding protein (IL-18BP), by modulating interferon regulatory factor 8 (IRF8) expression [57].

LXRs play significant roles in regulating immune responses in the liver. In LXRα/β knockout (LXRα/β-/-) mice, there is an increase in bone marrow-derived pro-inflammatory M1-marked F4/80+ CD11b+ KCs/macrophages, accompanied by elevated expression of inflammatory cytokines, leading to aggravated liver damage and heightened inflammatory responses [58]. Beyond repressing inflammatory gene expression, LXRs can also activate TLR-4 expression, modulating the innate immune response of macrophages [59]. In primary human macrophages, LXR ligands increase TLR-4 gene expression by enhancing the activity of TLR-4 promoters through binding to LXR response elements of the DR4-type (direct repeat with a spacing of four nucleotides) [59]. However, LXRs’ regulation of TLR-4 exhibits species specificity; in human macrophages, LXR agonists can activate LPS-induced TLR-4 expression, while this effect is not observed in mouse macrophages [59]. Additionally, the inhibitory effect of TLR3/4 on LXRs is mediated by IRF3, a specific effector of TLR3/4 that inhibits the transcriptional activity of LXRs on their target promoters [60]. Furthermore, LXRs are implicated in trained innate immunity; their activation induces a pro-inflammatory trained immunity phenotype in human monocytes, involving epigenetic and metabolic reprogramming linked to acetyl-CoA and IL-1β production [61]. The anti-inflammatory effects of LXR activation in monocyte-macrophages are summarized in Figure 2B.

LXRs significantly impact macrophage apoptosis and pathogen clearance. Bone marrow transplantation studies in mice have demonstrated that LXR-deficient mice are highly susceptible to Listeria infection. Moreover, macrophages lacking LXRs exhibit accelerated apoptosis following Listeria infection, resulting in diminished pathogen clearance capacity [62]. In mouse models of bacterial infection, the activation of TLRs suppresses anti-inflammatory LXR signaling, characterized by increased expression of LXRα and unaltered expression of LXRβ [23]. LXRα signaling also reduces Mycobacterium tuberculosis survival by altering cholesterol metabolism and promoting macrophage apoptosis [63]. Additionally, LXRs decrease macrophage capacity to clear Salmonella in a CD38-dependent manner [64]. LXR/Retinoid X receptor (RXR) heterodimer activation has been reported to inhibit bacterial pathogen-induced macrophage apoptosis. LXR and RXR agonists increase the expression of anti-apoptotic regulators such as AIM/CT2, Bcl-XL, and Birc1a while suppressing pro-apoptotic factors including caspases 1, 4, 11, 7, and 12, as well as the Fas ligand [65].

The phagocytic activity of macrophages, crucial for clearing apoptotic cells and maintaining immune homeostasis, is regulated by LXRs. Macrophages deficient in LXRs exhibit impaired phagocytic function and produce an abnormal pro-inflammatory response to apoptotic cells [66]. In lupus-like autoimmune mouse models, treatment with LXR agonists compensates for specific phagocytic defects in macrophages, improving disease progression [66]. Research indicates that LXR agonists in Ch25h-/- mice can mitigate LPS-induced neutrophilia, suggesting that LXRs facilitate neutrophil clearance by macrophages [67]. Additionally, 25-hydroxycholesterol (25HC) promotes the clearance of apoptotic neutrophils by monocyte-derived macrophages during the resolution of lung inflammation in an LXR-dependent manner [68]. LXRs also influence the phagocytic activity of liver macrophages against bacteria. In vitro studies show that LXR ligands enhance the phagocytic abilities of resident KCs, whereas LXRα knockout mice display diminished phagocytic activity against E. coli [53]. In human monocytes and macrophages, LXR activation increases retinoic acid receptor alpha (RARα) expression, which induces transglutaminase 2 (TGM2) expression, enhancing the phagocytosis of apoptotic cells [69]. Furthermore, LXRs boost macrophage phagocytic capacity by upregulating SREBP-c1, which increases retinaldehyde dehydrogenase 1 (RALDH1) expression, leading to retinoid production that activates PXR and upregulates phagocytic genes [70].

In conclusion, LXRs are critical regulators of macrophage function and immune homeostasis, influencing inflammatory, apoptosis, phagocytosis, and pathogen clearance, and offering potential therapeutic targets for enhancing immune regulation.

### 4.3. VDR Activation Modulates Immune Responses in Monocytes-Macrophages

VDR and its ligands play a crucial defensive role against bacterial pathogens. In VDR-deficient (VDR-/-) mouse models of chlamydial infection, bacterial clearance was impaired compared to wild-type mice [71]. In bone marrow-derived macrophages and monocytes, 1,25(OH)2D3 and its analogs induce the expression of the human cathelicidin antimicrobial peptide and defensin β2 (DEβ2) (Figure 2C) [71,72]. Early studies highlighted vitamin D’s potential in treating tuberculosis, showing that TLR activation in human macrophages upregulates VDR and Cyp27B1 expression, inducing cathelicidin production and facilitating the clearance of intracellular Mycobacterium tuberculosis [73,74].

Further research indicates that IL-1β is crucial for the upregulation of TLR2/1-induced DEFβ4 expression, necessitating the convergence of IL-1β and VDR pathways (Figure 2C) [75]. VDR complex binding to vitamin D response elements (VDRE) significantly enhances nucleotide-binding oligomerization domain 2 (NOD2) expression in epithelial and bone marrow cells [76]. NOD2, an intracellular pattern recognition receptor, bolsters host defense by stimulating the synthesis of antimicrobial peptides.

In a mouse model of acute hepatitis, VDR plays a significant role in liver immune cells. Compared to wild-type mice, VDR-/- mice show relatively alleviated Concanavalin A-induced hepatitis, with increased inflammatory cytokine gene expression, reduced reactive oxygen species (ROS) levels, and diminished phagocytic activity of KCs and hepatic neutrophils [77]. Additionally, 1,25(OH)2D3 enhances the transcription of ankyrin repeat and PH domain 2 (ASAP2) by binding to VDR-associated enhancer regions, thereby promoting macrophage phagocytosis [78]. The absence of VDR in macrophages leads to excessive production of miR-155, inhibiting suppressor of cytokine signaling 1 (SOCS1) and resulting in increased LPS-induced inflammatory responses [79]. VDR can also inhibit NF-κB activation by directly interacting with IKKβ [79]. The TLR2/1 signaling pathway mediated by VDR is influenced by multiple factors, including environmental, epigenetic, and genetic elements. The ultraviolet index can impact 25(OH)D3 levels, which in turn modulates VDR expression through methylation, enhancing VDR’s transactivation of collection of anti-microbial peptides (CAMP), which encodes the antimicrobial peptide hCAP-18 [80].

Overall, these findings underscore the versatile roles of VDR in modulating immune responses and highlight its importance as a target for therapeutic interventions against bacterial infections and inflammation.

### 4.4. GPBAR1 Activation Modulates Immune Responses in Monocytes-Macrophages

Kawamata et al. first described GPBAR1 as a bile acid receptor that plays a regulatory role in monocyte and macrophage activity [81]. Studies have shown that the cAMP-PKA-CREB pathway suppresses NF-kB activity [82]. Additionally, Yoneno et al. demonstrated that GPBAR1 agonists reduce the expression of pro-inflammatory cytokines like interferon-γ (IFN-γ), IL-1β, IL-6, and TNF-α, while increasing anti-inflammatory IL-10 expression in both human and mouse macrophages [83,84]. This effect is closely linked to increased IL-10 transcription, as IL-10-/- mice do not exhibit similar benefits [84]. Moreover, activation of GPBAR1 through an NF-kB-dependent mechanism reduces the accumulation of activated macrophages in atherosclerotic plaques and adipose tissues [85]. Using bone marrow chimeric and myeloid cell-specific GPBAR1 knockout mice, researchers found that selective GPBAR1 deletion impairs the protein kinase B (Akt)-mTORC1 pathway, leading to a pro-inflammatory phenotype [85]. GPBAR1 regulation also influences liver endothelial cell and monocyte-macrophage interactions by modulating CCL2/C-C motif chemokine receptor 2 (CCR2) transcription [86]. Additionally, BAR501 enhances IL-10 gene expression in liver and monocyte-macrophage cells, countering the inhibitory effects of acetaminophen (APAP) on this cytokine [86]. The interaction between GPBAR1 and CCL2/CCR2 involves the BAR501-mediated blockade of forkhead box O1 (FOXO1) binding, reducing Ccl2 and Ccr2 gene transcription [86].

Under pathological conditions, GPBAR1 influences macrophage function significantly. Activation of GPBAR1 can alleviate colitis in mice by altering macrophage polarization from the pro-inflammatory M1 phenotype to the anti-inflammatory M2 phenotype [84]. In contrast, GPBAR1-/- mice predominantly exhibit M1 macrophages, resulting in severe colitis when induced by TNBS [84]. GPBAR1 agonists reduce the presence of monocytes-macrophages in the gut and circulation without altering the ratio of resident to inflammatory monocytes, suggesting that monocyte differentiation depends on the organ microenvironment [87]. During hepatic ischemia-reperfusion injury, GPBAR1 modulates innate immune activation, with the GPBAR1-Cat E axis influencing pro-inflammatory responses by targeting macrophage motility and polarization. GPBAR1 deficiency increases CD11b+ macrophage infiltration, whereas INT-777 reduces macrophage accumulation in wild-type mice [88]. Reduced GPBAR1 expression has been noted in non-alcoholic steatohepatitis models, where GPBAR1 inhibits NLRP3 activation and fosters macrophage M2 polarization to mitigate inflammation [89]. Interestingly, the FXR/GPBAR1 dual agonist INT-767 mitigates liver damage, improves histology, increases M2 macrophage markers, and decreases Ly6C monocyte ratios in obese db/db mice, promoting an anti-inflammatory phenotype [90]. These effects were confirmed by in vitro observations, where INT-767 treatment reduced Ly6C expression and increased IL-10 production via the cAMP pathway [90]. GPBAR1 selective ligands such as BAR501 or dual ligands like BAR502 improve steatosis and fibrosis indices in high-fat diet models by enhancing macrophage polarization towards the M2 phenotype, reducing pro-inflammatory markers, and alleviating hepatic fat deposition [91]. Similar results have been corroborated using additional FXR/GPBAR1 dual ligands [92].

This comprehensive body of research highlights GPBAR1′s crucial role in modulating immune responses (Figure 2D), offering potential therapeutic pathways for inflammatory diseases and metabolic disorders.

## 5. Roles of BA-Activated Receptors in DCs

### 5.1. FXR and GPBAR1 Activation Modulates the Inflammation in DCs

In addition to monocytes and macrophages, the expression of FXR and GPBAR1 has been identified in DCs [21,93]. DCs serve as crucial sentinels of intestinal epithelial cells, sensing pathogens and guiding appropriate immune responses to maintain tissue homeostasis while acquiring functional phenotypes that can either enhance or mitigate inflammatory responses [52,94]. Gadaleta et al. (2011) demonstrated that activation of FXR by the selective agonist INT-747 can reduce the severity of DSS and TNBS-induced colitis in mice, leading to decreased production of the pro-inflammatory cytokine TNF-α in DCs [50]. Similarly, Massafra et al. (2016) highlighted the beneficial effects of FXR activation on DSS-induced colitis, showing that OCA activation promotes an anti-inflammatory state by increasing the retention of DCs in the spleen, thereby decreasing their presence in the colon and alleviating colonic inflammation [52].

Ichikawa et al. explored the role of GPBAR1 in DCs and found that activation of GPBAR1 induces lower production of IL-12 and TNF-α in response to bacterial antigens. Furthermore, the presence of GPBAR1 agonists promotes a low-secreting phenotype of IL-12 during DC differentiation [94]. DCA, as a key regulatory factor in the pathogenesis of autoimmune uveitis, activates GPBAR1, inhibits NF-κB activation in DCs via the cAMP-PKA pathway, and reduces the production of pro-inflammatory cytokines such as IL-1β, IL-12p70, TNF-α, and IL-6 (Figure 2E) [93]. While GPBAR1 inhibits NF-κB activation, the phosphorylation of CREB and the secretion of IL-10 do not appear to be involved in this process [93]. Additionally, research has shown that both DCA and INT-777 can suppress the expression of pro-inflammatory cytokines and surface markers in monocyte-derived dendritic cells (Mo-DCs) from patients with Behçet’s disease and Vogt–Koyanagi-Harada syndrome, while concurrently reducing NF-κB activation [93].

These findings underscore the critical roles of GPBAR1 and FXR in regulating dendritic cell function, suggesting their potential as therapeutic targets for modulating immune responses in various inflammatory and autoimmune conditions.

### 5.2. LXR Activation Regulates Inflammation and the Immune Response in DCs

LXRs expressed in DCs play a significant role in maintaining neutrophil balance within the body [95]. Effective clearance of senescent neutrophils in mice relies on LXR signaling activated by the phagocytosis of apoptotic cells [95]. Both natural and synthetic LXR agonists have been shown to promote DC maturation and enhance the production of pro-inflammatory cytokines such as IL-12, TNF-α, IL-6, and IL-8, thereby improving the ability of DCs to induce CD4+ T cell proliferation [95]. However, LXR agonists can also inhibit CCR7 expression and DC migration in both in vitro and in vivo settings [95]. Furthermore, studies demonstrate that LXRs can interact directly with the NF-κB subunit p50 in DCs, preventing its translocation into the nucleus. This interaction inhibits NF-κB activation and reduces the production of cytokines such as IL-12, IL-23, and IL-27 (Figure 2F) [96]. Activation of LXRs enhances DC maturation at the phenotypic, cytokine, and functional levels [97]. Additionally, treating human DCs with LXR agonists can amplify the expression of inflammatory cytokines following stimulation with TLR-3 or TLR-4 agonists [23]. Plasmacytoid dendritic cells (pDCs), which appear in atherosclerotic lesions, may also play a role in regulating atherosclerosis. Stimulation of the LXR pathway in pDCs inhibits TLR7-induced NF-κB activation and secretion of pro-inflammatory cytokines while enhancing microparticle internalization through brain-specific angiogenesis inhibitor 1 [98]. These findings highlight the dual roles of LXRs in immune regulation, impacting both inflammation and the immune response during pathological conditions.

### 5.3. VDR Activation Regulates the Immune Response in DCs

The VDR and its ligands play a critical role in regulating the maturation, differentiation, and immune functions of DCs [95,99]. The VDR ligand 1,25(OH)2D3 has been shown to inhibit the differentiation and maturation of DCs in vitro [100], and specifically, 1α,25-(OH)2D3 suppresses DC maturation through VDR [101]. Studies in mice demonstrate that VDR-deficient mice exhibit hypertrophy of subcutaneous lymph nodes with an increased population of mature DCs, while the number of splenic DCs remains unchanged compared to wild-type mice [101]. In addition to its role in DC differentiation and maturation, 1,25(OH)2D3 modulates the expression of chemokines in myeloid DCs (M-DCs). Notably, it upregulates the production of CCL22 and inhibits the production of CCL17 [102]. Furthermore, 1,25(OH)2D3 suppresses the phosphorylation and nuclear translocation of NF-κB p65 in M-DCs, indicating a mechanism through which it regulates inflammatory responses (Figure 2F) [102]. VDR gene polymorphisms can also influence immune cell functions, with shorter F-VDR variants being associated with a more active immune response [103]. These findings underscore the multifaceted roles of VDR in immune regulation, highlighting its importance in both dendritic cell function and mucosal immunity.

## 6. Conclusions and Future Perspective

Bile acid-activated receptors, including FXR, GPBAR1, LXR, and VDR, play pivotal roles in modulating immune responses within monocytes-macrophages, and DCs. FXR and GPBAR1 significantly contribute to dampening inflammatory responses and promoting anti-inflammatory macrophage phenotypes. FXR inhibits the production of pro-inflammatory cytokines and helps regulate chemokine environments, thereby influencing DC function and differentiation. Similarly, GPBAR1 reduces cytokine production via modulation of the NF-κB pathway-albeit through distinct mechanisms from FXR and promotes an anti-inflammatory state in both macrophages and DCs. LXRs are critical for regulating lipid metabolism and inflammatory pathways, controlling cytokine production and enhancing phagocytic activities, which in turn manage DC maturation, function, and macrophage polarization. Meanwhile, VDR plays a crucial role in immunomodulation by facilitating the differentiation and maturation of DCs and macrophages, influencing chemokine expression, and inhibiting NF-κB signaling. Collectively, these receptors coordinate innate immune responses, maintaining immune homeostasis while presenting potential therapeutic targets for managing inflammatory and autoimmune diseases (Figure 3).

Future research should aim to elucidate the intricate signaling pathways through which these receptors exert their effects on monocytes, macrophages, and DCs. Deeper insights into receptor interactions and their impacts on cellular function could clarify their roles in disease contexts such as autoimmunity, chronic inflammation, and metabolic disorders. Furthermore, the development of receptor-specific agonists or antagonists presents a promising therapeutic avenue. Investigating how these treatments can modulate immune cell functions without compromising systemic immune competence will be crucial. The role of genetic variations, such as polymorphisms in VDR, may also facilitate personalized therapeutic approaches tailored to individual genetic makeups. Finally, exploring the interplay between these receptors and the microbiome could reveal novel strategies for enhancing mucosal immunity and overall immune resilience. These efforts will advance our understanding of immune regulation and foster the development of targeted interventions for immune-mediated diseases.

## Figures and Tables

**Figure 1 cells-14-00920-f001:**
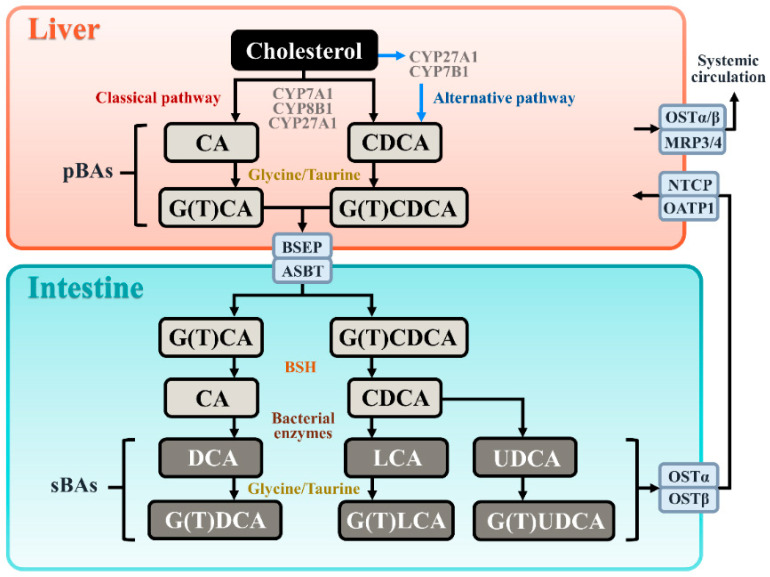
Biosynthesis and metabolism of BAs. This schematic illustrates the synthetic pathways of primary BAs in hepatocytes (depicted in orange) and secondary BAs in the intestine (depicted in blue). CA, cholic acid. DCA, deoxycholic acid. CDCA, chenodeoxycholic acid. LCA, lithocholic acid. UDCA, ursodeoxycholic acid, BSEP, bile salt export pump. ASBT, apical sodium-dependent bile acid transporter. OSTα/β, organic solute transporter α/β. MRP3/4, multidrug resistance-associated protein 3/4. NTCP, sodium/taurocholate co-transporting polypeptide. OATP1, organic anion-transporting polypeptide 1.

**Figure 2 cells-14-00920-f002:**
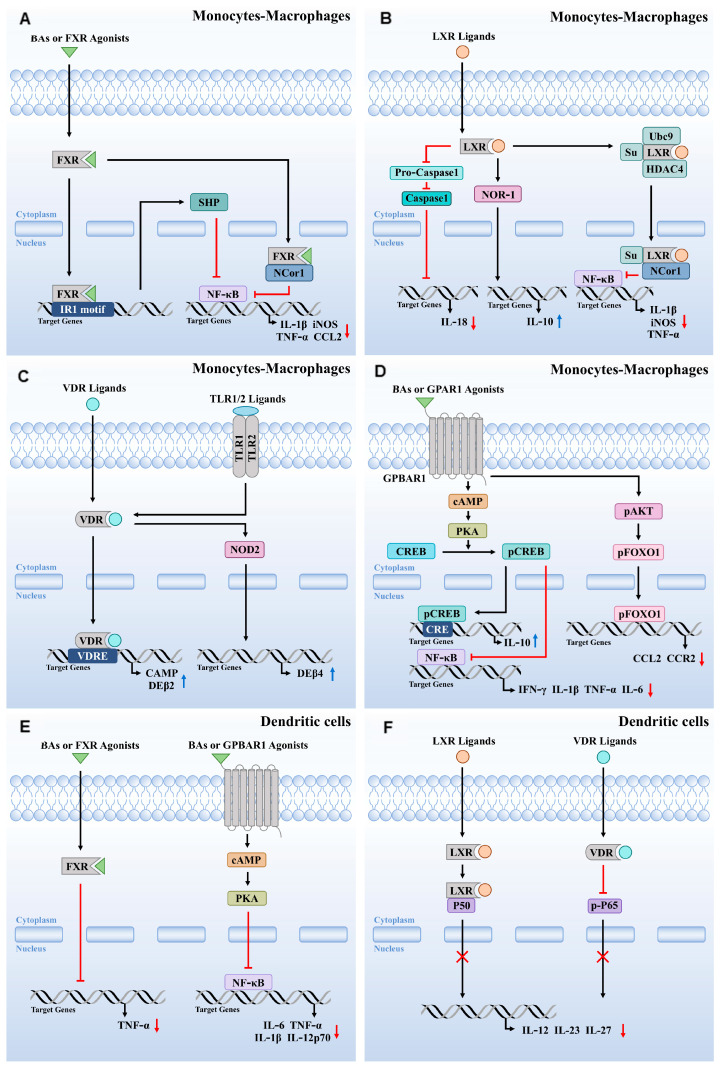
The immune response modulating effects of FXR, LXR, VDR, and GPBAR1 activation in monocytes-macrophages and DCs. (**A**) FXR activation leads to SHP or FXR-NCor1 mediated repression of NF-κB and inhibit expression of IL-1β, iNOS, TNF-α, and CCL2 in monocytes-macrophage. (**B**) The anti-inflammatory effects of LXR activation in monocyte-macrophages. (**C**) VDR signaling leads to VDR and VDR-NOD2-mediated expression of CAMP, DEβ2 and DEβ4, respectively. (**D**) The immune response modulating effects of GPBAR1 activation in monocyte-macrophages. (**E**) The anti-inflammatory effects of FXR and GPBAR1 activation in DCs. (**F**) LXR and VDR activation leading to P50 or p-P65 mediated repression of IL-12, IL23, and IL-27. SHP, small heterodimer partner. NF-κB, nuclear factor-kappa B. NCor1, nuclear receptor corepressor 1. IL-1β, interleukin-1β. iNOS, inducible nitric oxide synthase. TNF-α, tumor necrosis factor-alpha. CCL2, C-C motif chemokine ligand 2. NOR-1, nuclear receptor orphan receptor 1. Ubc9, ubiquitin conjugating enzyme 9. HDAC4, histone deacetylase 4. IL-18, interleukin-18. COX-2, cyclooxygenase-2. IL-10, interleukin-10. VDRE, vitamin D response elements. NOD2, nucleotide-binding oligomerization domain 2. TLR1/2, toll-like receptor 1/2. CAMP, collection of anti-microbial peptides. DEβ2/4, defensin β2/4. cAMP, cyclic adenosine monophosphate. PKA, protein kinase A. CREB, cyclic-AMP response element-binding protein. CRE, cAMP response element. AKT, protein kinase B. FOXO1, forkhead box O1. IL-6, interleukin-6. IFN-γ, interferon-γ. CCR2, C-C motif chemokine receptor 2. IL-12, interleukin-12. IL-23, interleukin-23. IL-27, interleukin-27. Arrows indicate activation (→) or inhibition (⊣). Blue arrows (↑) indicate up-regulation, red arrows (↓) indicate down-regulation, and red cross (×) indicate inhibition.

**Figure 3 cells-14-00920-f003:**
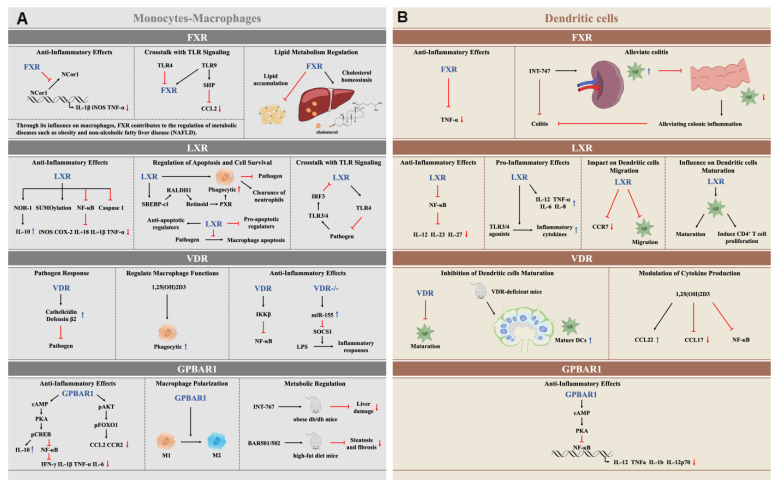
Summarizes the immune-modulating effects of bile acid-activated receptors (FXR, GPBAR1, LXR, and VDR) in monocytes-macrophages (Panel (**A**)) and dendritic cells (DCs, Panel (**B**)). Each subpanel illustrates the key signaling pathways and outcomes of receptor activation, as detailed in the text. Panel A highlights the anti-inflammatory roles of FXR and GPBAR1, the dual regulation of inflammation and lipid metabolism by LXR, and the antimicrobial and immunomodulatory effects of VDR in monocytes-macrophages. Panel B depicts the receptor-specific modulation of DC functions, including cytokine production, maturation, and NF-κB inhibition. Arrows indicate activation (→) or inhibition (⊣). Blue arrows (↑) indicate up-regulation and red arrows (↓) indicate down-regulation. For further details, refer to Section 4 and Section 5.

**Table 1 cells-14-00920-t001:** Distribution, natural ligands, and antagonists of BA-activated receptors.

BA-Activated Receptors		Tissue Distributions	Cell Distributions	BA Ligands	Antagonists	References
Nuclear receptors	FXR	liver, heart, kidneys, intestines	hepatocytes, cholangiocytes, liver sinusoidal cells, intestinal and liver endothelial cells, monocytes-macrophages, DCs, NK cells, NKT cells, T cells, B cells	CDCA > LCA > DCA > CA	UDCA, β-MCA, Gly-MCA	[17,18,46]
	LXR	intestines, liver, kidneys, lungs, adipose tissue	hepatocytes, macrophages	6α-hydroxylated BAs	-	[22,27]
	VDR	ileum, kidneys, liver, bones, skin, endocrine tissues	T cells, B cells, NK cells, DCs, monocytes-macrophages, non-parenchymal liver cells	DCA, LCA	-	[30,31]
	PXR	liver, intestines	hepatocytes	3-keto-LCA, LCA, DCA	-	[36]
	CAR	liver, small intestine, gallbladder	hepatocytes	CA, LCA	-	[37]
Cell membrane receptors	GPBAR1 (TGR5)	intestines, stomach, liver, spleen, gallbladder, adipose tissue	sinusoidal endothelial cells, cholangiocytes, hepatic stellate cells, monocytes-macrophages, DCs, NKs cells, NKT cells	TLCA > LCA > DCA > CDCA > CA	-	[33,46]
	S1PR2	Most tissues	hepatocytes	Taurine and glycine conjugated BAs	-	[38,47]
	M2 and M3	heart, central nervous system	smooth muscle cells	DCA-LCA	-	[6]
	FPR1	-	macrophages	-	DCA, CDCA	[39]

Note: “-” means not reported.

## Data Availability

No new data were created or analyzed in this study.

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
