# Peer review of "Roles of Bile Acid-Activated Receptors in Monocytes-Macrophages and Dendritic Cells"

_cells, 2025, doi:10.3390/cells14120920_

Round 1

Reviewer 1 Report

Comments and Suggestions for Authors

This is a comprehensive review of the incidence and interactions of bile acid receptors in non-hepatic inflammatory cells, including macrohages,dendridic cells  and others.

While these interactions are complex,I would have liked to see some physiologic  and even disease phenotypyes that provided examples of these interactions rather than the general implication that they resulted in roles in inflammation.

Minor suggestions: 1) On p3.liine 85, please add Ostalpha/beta as an additional bile acid transporter that functions as both an uptake and an excretory transporter on the basolateral domain of the hepatocyte.

2) What are the effects of elevated levels of bile acids on some of these these pathways?e.g.cholestasis

3). Aside from surface receptors, How do bile acid get into these inflammatory cells to interact with intracellular nuclear receptors? Please discuss.

3) 

Reviewer 2 Report

Comments and Suggestions for Authors

The manuscript reviews the role of bile acids mediated nuclear receptor activated immune responses. It covers very significant aspects especially the role of Bile acids in down regulation of immune responses with reference to Monocytes-Macro- 2 phages and Dendritic Cells.

The manuscript seems written well. However, some aspects in describing the down regulation of NF-kB is lacking. This is pointed below.

One of the major concerns of the manuscript is not describing the “Nuclear Cross-talk of NF-kB with other members of the nuclear receptor superfamily. In addition, the manuscript did not explore to delineate the role of Activator Protein-1 (AP-1) which is also an inflammatory transcription factor in immune function. It is recommended that authors explore to define nuclear cross cross-talk of LXR, FXR, and VDR(activated by Bile acids)  with NF-kB.

In addition Nuclear co-repressor (NCoR) does not possess binding to DNA but associates as co-regulator (as inhibitor) which should be defined correctly.

Line 216-217/217: NCoR does not bind to DNA as mentioned for binding to response element in the sentence “This binding stabilizes the nuclear receptor corepressor 1 (NCoR1) complex, which associates with the NF-κB response element (Figure 2A)” and Figure 2B (showing the arrow

Line 197: SHP is not written in full. SHP full name is Small heterodimer partner (SHP) which lacks the conserved DNA-binding domain, hence when heterodimerized will reduce the transcription.

Many other transcription factors’ name such as RXR and Ch2h are not written in full.

Line 253: The word “DR4: in the sentence “………o DR4-type LXR response elements.” Is not defined. The NR members bind with nucleotide spacing, here a spacing of FOUR NUCLEOTIDES between the direct repeat. It should written” well” here.

Reviewer 3 Report

Comments and Suggestions for Authors

1. Recommend redesigning Table 1, the current format is hard to read.

2. Figure 3 legend needs more details.

3. Consider adding subtitles to highlight the role of BAARs in myeloid cells in human disease, such as liver disease, metabolic syndrome, IBD, etc. 

4. English editing is strongly recommended to avoid confusions. 

Comments on the Quality of English Language

English editing is strongly recommended. 

Reviewer 4 Report

Comments and Suggestions for Authors

Jia H. et al in their review on bile acid receptors in macrophages and dendritic cells give a nice overview of the bile acid-activated receptors, their signaling mechanisms and implications. It is a timely review to understand the pleiotropic role these receptors play in physiology and pathobiology.

My only comment would be to add few statements on the known role of these receptors in human diseases. The section where the authors introduce all the bile-activated receptors, what is known about these receptors in liver and intestinal disorders can be of interest to the readership.

Round 2

Reviewer 1 Report

Comments and Suggestions for Authors

Please add references to back up your added statement concerning the number of  bile acid transporters that you say are present on inflammatory/immune cells. Only ref 11 is added.

Reviewer 2 Report

Comments and Suggestions for Authors

The modifications and corrections have improved the quality of manuscript.

Author Response

Thanks so much for your positive comments!

Reviewer 3 Report

Comments and Suggestions for Authors

The authors didn't address comment 3. 
